# Microbial Diversity and Characteristic Quality Formation of Qingzhuan Tea as Revealed by Metagenomic and Metabolomic Analysis during Pile Fermentation

**DOI:** 10.3390/foods12193537

**Published:** 2023-09-22

**Authors:** Lin Feng, Shiwei Gao, Panpan Liu, Shengpeng Wang, Lin Zheng, Xueping Wang, Jing Teng, Fei Ye, Anhui Gui, Jinjin Xue, Pengcheng Zheng

**Affiliations:** 1Fruit and Tea Research Institute, Hubei Academy of Agricultural Sciences, Wuhan 430064, China; fenglin@hbaas.com (L.F.); gsw0609@126.com (S.G.); liuppitea@163.com (P.L.); wwsspp0426@163.com (S.W.); caozi20121117@163.com (L.Z.); wangxueping79-79@163.com (X.W.); jobbase@163.com (J.T.); yf421@163.com (F.Y.); guianhui@tricaas.com (A.G.); xuejinjin911@163.com (J.X.); 2Key Laboratory of Tea Resources Comprehensive Utilization, Ministry of Agriculture and Rural Affairs, Hubei Tea Engineering and Technology Research Centre, Wuhan 430064, China

**Keywords:** Qingzhuan tea, pile fermentation, microorganisms, quality-related components

## Abstract

In order to analyze the changes in the microbial community structure during the pile fermentation of Qingzhuan tea and their correlation with the formation of quality compounds in Qingzhuan tea, this study carried out metagenomic and metabolomic analyses of tea samples during the fermentation process of Qingzhuan tea. The changes in the expression and abundance of microorganisms during the pile fermentation were investigated through metagenomic assays. During the processing of Qingzhuan tea, there is a transition from a bacterial dominated ecosystem to an ecosystem enriched with fungi. The correlation analyses of metagenomics and metabolomics showed that amino acids and polyphenol metabolites with relatively simple structures exhibited a significant negative correlation with target microorganisms, while the structurally complicated B-ring dihydroxy puerin, B-ring trihydroxy galloyl puerlin, and other compounds showed a significant positive correlation with target microorganisms. *Aspergillus niger*, *Aspergillus glaucus*, *Penicillium* in the *Aspergillaceae* family, and *Talaromyces* and *Rasamsonia emersonii* in *Trichocomaceae* were the key microorganisms involved in the formation of the characteristic qualities of Qingzhuan tea.

## 1. Introduction

Pile fermentation is a key process in the formation of the characteristic quality of dark tea. This process is characterized by the growth and succession of microbial communities [1,2,3]. During the pile fermentation process, microorganisms secrete extracellular enzymes to promote the conversion of chemical compounds in tea leaves. The decomposition of protein, pectin, and cellulose, and the enzymatic oxidation of catechins all occur with the help of microorganisms [4]. Many functional core microorganisms such as *Aspergillus*, *Cyberlindnera*, *Rasamsonia*, *Bacillus*, *Debaryomyces*, *Aspergillus*_*fumigatus*, and *Bacillus*_*subtilis* participate in the formation of the main quality components of dark tea [5,6,7]. During microbial fermentation, the contents of catechins, flavonoids, alkaloids, and amino acids are usually greatly reduced, while the contents of theabrownin (TB) and phenolic acids are increased. The changes in these quality compounds are mainly due to reactions such as degradation, oxidation, condensation, structural modification, methylation, and glycosylation induced by microbial extracellular enzyme catalysis or microbial metabolism [1,6,8,9]. The transformation of dark tea catechins is mainly a degradation reaction and it can lead to the hydrolysis of galloyl catechins under the action of microbial extracellular enzymes, thereby producing (-)-gallocatechin and gallic acid [1]. Many phenolic acids are formed during the fermentation of dark tea including 2, 5-dihydroxyphenylcarboxylic acid, phloroglucinol, salicylic acid, protocatechuic acid, and others. These phenolic acids are also produced from the transformation of microorganisms [10,11]. Theabrownin is mainly formed through oxidative polymerization of polyphenols (mainly composed of catechins) under the conditions of microbial fermentation and microbial extracellular enzyme interactions (such as polyphenol oxidase, peroxidase, pectinase, and others) [12].

The main microbial genera in the pile fermentation of dark tea include *Aspergillus*, *Penicillium*, *Debaryomyces*, *Bacillus, Lactococcus*, and *Enterobacter*, and others [13,14]. *Aspergillus* has been identified as the dominant microorganism in the early stage of pile fermentation of Pu’er tea and it is mainly involved in multiple biological processes such as the transformation from tea polyphenols to theabrownin and the degradation of caffeine [15,16]. Moreover, several enzymes secreted from the *Aspergillus* genus, such as cellulases, hemicellulases, proteases, and α-amylases, could catalyze the major metabolites in Pu’er tea during the fermentation process [17]. The fungal genus *Aspergillus* is considered to be the most prevalent genus during the microbial fermentation of Pu’er tea, while *Aspergillus niger* has been often documented as the most dominant species [8,18]. Compared with the fungal community, the bacterial community in mature Pu’er tea has been less studied, and some bacterial genera have been identified, including *Pseudomonas, Bacillus*, and *Brevibacterium*, and a variety of thermophilic bacterial species have been isolated during microbial culture [19]. Fungal and bacterial communities are the dominant strains in Fuzhuan tea. *Firmicutes* and *Proteobacteria* are the dominant bacterial phyla. *Ascomycota, Trichomycetes*, and *Aspergillus* are the dominant fungi, and fungi have been reported to make greater contributions to tea polyphenol metabolism compound conversion and tea quality formation [1,5,6,20,21]. The metagenomic method can be used to directly sequence and analyze the total DNA in microbial combinations. It can effectively avoid the inherent bias caused by PCR and primer selection, and thus it can accurately reveal the types, quantities, and proportions of different microorganisms in the community. It can also mine the metabolic potentials of all microorganism communities [14,22,23]. In recent years, the application of metagenomics in food, agriculture, the environment, and other fields has increased significantly. However, there have been few reports on the microorganism changes and the identification of dominant strains during the fermentation process of Qingzhuan tea. *Aspergillus* has been regarded as the predominant genus in the pile fermentation of Qingzhuan tea, followed by the *Penicillium*, *Debaryomyces*, *Thermomyces*, *Rasamsonia*, and *Byssochlamys* genera [24,25]. However, a series of scientific issues such as the dominant microorganisms in Qingzhuan tea and their contributions to the formation of tea quality remain largely unclear. In recent years, the metabolomic method has been widely used for the exploration of the formation mechanism of tea quality, functional components, and the changes in compound fingerprint profiles before and after drinking tea [26,27,28]. Xu et al. (2015) used LC-MS to analyze the changes in compounds during the fermentation process of Fuzhuan tea, and found great changes in important metabolites affecting the Fuzhuan tea quality such as caffeine, catechin, and gallic acid [29]. Xiao et al. (2022) used UHPLC/Q-TOF-MS technology to analyze the metabolomes of dark tea with different fermentation durations, and they found that the metabolites of dark tea changed gradually during the fermentation process and significant changes occurred in the phenolic metabolism synthesis pathway [27]. However, the changes in compounds and the effects of the interaction between compounds and microorganisms on tea quality during the fermentation of Qingzhuan tea remain largely unknown.

In this study, the metagenomic method was used to analyze the dynamic changes in the microbial community structure during the pile fermentation stage of Qingzhuan tea and metabolomics was employed to investigate the dynamic changes in the quality compounds of Qingzhuan tea. This study aimed to explore the correlation between the dominant strains and key quality compounds during the pile fermentation process and to reveal the effect of microorganisms on the transformation and formation of key metabolites. Our findings will provide new perspectives for analyzing the quality formation mechanism of Qingzhuan tea.

## 2. Materials and Methods

### 2.1. Experimental Materials

The experimental samples used in this study were obtained from the Zhaoliqiao Tea Factory in Chibi city, Hubei province, China. Sampling was started at the beginning of the pile fermentation of raw tea, and sampling was conducted every other day 10 times during entire pile fermentation. The samples from the first sampling were named QZT1, and the samples obtained from the 1st to the 10th were successively named QZT1~QZT10. The fermentation piles were turned three times. QZT1~QZT4 were the samples obtained from the first pile turning, QZT5~QZT7 from the second pile turning, and QZT8~QZT10 from the third pile turning. Each sample was randomly selected from 10 points in different layers of the large fermentation pile. About 1 kg of the sample was mixed, sealed in a sterile sampling bag, and quickly frozen on dry ice for metagenomic and metabolomic sequencing analyses. The LC-MC analysis and metagenomic analysis were conducted with three biological replicates.

### 2.2. Experimental Methods

#### 2.2.1. CTAB Genomic DNA Extraction, Library Construction, and Sequencing

The 60 g tea samples were added into 800 mL sterile water and stirred thoroughly to obtain a mixture solution. After filtering with sterile gauze, the filtrate was centrifuged at 1200 rpm for 10 min at 4 °C. Afterwards, the supernatant was collected and centrifuged at 10,000 rpm for 10 min, and then the precipitate was collected and centrifuged twice at 120,000 rpm for 10 min each time, followed by DNA extraction.

DNA was extracted using the CTAB (cetyltrimethylammonium bromide) method with the following steps. (1) The 1000 μL CTAB lysate was sucked into a 2.0 mL EP tube. An amount of 20 μL of lysozyme was added and an appropriate amount of tea sample was added into the lysate. The samples were subjected to a water bath at 65 °C, inversion mixed evenly, and lysed thoroughly. (2) After centrifugation, 950 μL supernatant was supplemented with an equal volume of the mixture of phenol (pH = 8.0), chloroform, and isoamyl alcohol (at the ratio of 25:24:1), inversion mixed, and centrifuged at 12,000 rpm for 10 min. (3) The supernatant was collected, supplemented with an equal volume of chloroform: isoamyl alcohol (24:1), inversion mixed, and centrifuged at 12,000 rpm for 10 min. (4) The supernatant was sucked into a 1.5 mL centrifuge tube, supplemented with 3/4 (supernatant) volume of isopropanol, shaken up and down, and precipitated at −20 °C for 5 min. (5) After centrifugation at 12,000 rpm for 10 min, the supernatant was removed and the precipitate was washed twice with 1 mL of 75% ethanol, and the remaining small amount of liquid was collected and centrifuged again, and then sucked out with a pipette. (6) The obtained precipitate was blow-dried on an ultra-clean bench and supplemented with 50 μL ddH_2_O to dissolve the DNA sample. (7) An amount of 1 μL RNase A was added to digest RNA. The sample solution stood at 37 °C for 15 min and was stored at −20 °C.

#### 2.2.2. Library Construction

An amount of 1 μg of genomic DNA was used to construct a metagenomic library. Specifically, the library was constructed using the NEBNext Ultra DNA Library Prep Kit for Illumina, and library construction steps included sample fragmentation, end repair and A-tailing addition, adapter connection, and library PCR amplification. Subsequently, PCR reaction product was purified with 0.6 times Agencourt AMPure XP (nucleic acid purification magnetic beads) and the concentration of the library was detected by Qubit2.0. Gene sequencing was conducted using Illumina PE150 from Nuohezhiyuan Company (Beijing, China).

#### 2.2.3. Metagenomic Data Assembly and Splicing

The Readfq (V8, https://www.github.com/cjfield/readfq, accessed on 4 July 2022) was used to preprocess the raw data obtained from the Illumina HiSeq sequencing platform to obtain clean data for subsequent analysis. SOAPdenovo V. 2.04 software (http://soap.genomics.org.cn/soapdenovo.html, accessed on 4 July 2022) was used to analysis the genome assembly based on the obtained clean data.

#### 2.2.4. Metagenome Gene Prediction

MetaGeneMark (V. 2.10, http://topaz.gatech.edu/GeneMark, accessed on 5 July 2022) was the platform used to predict the open reading frame (ORF) of the scaffolds of the samples and information with a length < 100 nt was removed from the prediction results. CD-HIT software (V. 4.5.8, http://www.bioinformatics.org/cd-hit/, accessed on 7 July 2022) was used to remove redundancy from ORF prediction results so as to obtain a non-redundant initial gene catalogue. Then, Bowtie 2 software (Bowtie 2.2.4) was used to align the clean data of the samples with their scaffolds. Based on the abundance information in the gene catalog, core–pan gene analysis, correlation analysis between samples, and gene number Venn diagram analysis were performed.

#### 2.2.5. Metagenomic Pathway Annotation

DIAMOND software (V0.9.9.110, http://github.com/bbuchfink/diamond/, accessed on 13 July 2022) was used to align the unigenes to the functional databases including the KEGG database (http://www.kegg.jp/kegg/, accessed on 14 July 2022), eggNOG (http://eggnogdb.embl.de/, accessed on 14 July 2022), and CAZy 24 151 (http://www.cazy.org/, accessed on 15 July 2022). The best blast hit result was selected from the alignment results of each sequence for subsequent analysis. Based on the abundance table at each taxonomic level, annotated gene number was counted; relative abundance overview was displayed; abundance cluster heatmap was plotted; and metabolic pathway comparative analysis, PCA (principal component analysis), NMDS (non-metric multidimensional scaling) dimensionality reduction analysis, and LEfSe (linear discriminant analysis effect size) analysis of functional differences between groups (with LDA score set as 4) were performed.

#### 2.2.6. Metabolomics Analysis

The tea samples were vacuum freeze-dried and then ground into powder. The 50 mg tea sample was added into 800 μL of pre-cooled methanol, vortexed for 1 min, ultrasonically extracted at 25 °C for 30 min, vortexed for 1 min, and centrifuged at 15,000 rpm for 15 min at 4 °C. The 200 μL of supernatant and 10 μL of inner standard (0.3 mg/mL, DL-4-chlorophenylalanine methanol solution) were put into the vial. The samples from each group were subjected to LC-Orbitrap-MS, and 10 μL samples from each group were mixed to establish a quality control sample (QC) for instrument stability monitoring and data deviation correction.

#### 2.2.7. Analysis Conditions of Liquid Chromatography and Mass Spectrometry

The liquid chromatography analysis conditions were as follows: Hypersil GOLD C18 (100 × 2.1 mm, 1.9 μm) column was adopted; the column temperature was 40 °C; the sample plate temperature was 4 °C; the mobile phase A was water +0.1% formic acid; the mobile phase B was acetonitrile +0.1% formic acid; the flow rate was 0.3 mL/min; and the injection volume was 2 μL. The gradient elution conditions are shown in Appendix A.

Mass spectrometry analysis conditions were as follows: Q-Extractive Focus high-resolution combined mass spectrometer was adopted and the mass spectrometer was equipped with a heatable electrospray ion source (HESI-II). The full-scan mode settings were as follows: For data collection in positive ion mode, the sheath gas flow rate was 45 arb; the auxiliary gas flow rate was 15 arb; the exhaust gas flow rate was 1 arb; the capillary temperature was 350 °C; the capillary voltage was 3.8 kv; and the RF (radio-frequency) energy level of the lens was 60%. For Controid data acquisition mode, the mass range of the acquisition was 50~1000 *m*/*z*; the resolution was 60,000, the number of data-dependent scanning triggers was 10, and the fragmentation voltage was 20 V ± 50%.

#### 2.2.8. Data Matrix Extraction

The chemical component data of each sample were collected using Xcalibur 2.1.x software, and the raw data in .RAW format were inputted into Compound Discoverer 2.0 software for peak identification and peak integration. Subsequently, retention time correction, peak alignment, background subtraction, and devolution were performed to obtain two-dimensional data matrix with the peak area, retention time, mass/charge ratio, and sample name of each group.

#### 2.2.9. Characterization of Compounds

By using the OPLS-DA (orthogonal partial least squares–discriminant analysis) model, the key difference compounds of samples in different processing procedures were analyzed. After an accurate mass/charge ratio list (primary mass spectrometry information) was obtained from the raw data matrix, the relevant secondary mass spectrometry information was obtained using LC-Orbitrap-MS. According to the primary mass spectrum information of differential compounds, the possible compounds were retrieved and listed by searching online databases Metlin (metlin.scripps.edu, accessed on 13 September 2022), HMDB (www.hmdb.ca, accessed on 20 September 2022), and KEGG (www.kegg.jp, accessed on 24 September 2022). According to the obtained secondary mass spectrum information of possible compounds and retention time, the secondary spectra of the corresponding compounds were searched in Metlin (www.metlin.scripps.edu, accessed on 22 September 2022), Mzcloud (www.mzcloud.org, accessed on 23 September 2022), and Massban (www.massbank.jp, accessed on 24 September 2022). According to the fragmentation energy, the corresponding fragment ions were aligned and the candidate compounds were further listed. Based on the retrieval information from TMDB (pcsb.ahau.edu.cn:8080/TCDB/f) database and the related literature, the candidate compounds were screened and characterized.

#### 2.2.10. Data Analysis

SPSS 17.0 software was used for variance analysis and correlation analysis. SIMCA-P 13 software was used for principal component analysis (PCA) and orthogonal partial least squares–discriminant analysis (OPLS-DA).

## 3. Results

### 3.1. Microbial Sequencing and Main Data Analysis during Qingzhuan Tea Pile Fermentation Process

Samples with three replicates from the pile fermentation process were used as the experimental materials. In total, 570,421.25 Mbp of raw data were obtained; after removing reads with low-quality bases (mass value <= 38), those with a lower proportion of N bases (default was set to 10 bp), and those that exceeded the threshold (default was set to 15 bp), 567,343.62 Mbp of clean data were obtained from sequencing by Illumina HiSeq. After assembly, a total of 6,682,219,438 bp of scaffolds were obtained. After gene prediction by MetaGeneMark Linux 64 software, a total of 6,353,703 ORFs were obtained. After redundancy removal, a total of 740,943 ORFs were obtained with a total length of 324.59 Mbp, of which the number of complete genes was 368,400, accounting for 49.72%.

Based on the abundance table of genes in each sample, the gene number information of each sample was obtained. Different numbers of samples were randomly selected to obtain the number of genes in the combinations of different numbers of samples, based on which the dilution curves of the core and pan genes were constructed and plotted (Appendix A). The closer the curve is, the smoother it becomes, indicating that, as the number of sequencing samples increases, the number of genes gradually stabilizes. The flat curve meant that the number of genes gradually tended to be stable with the increasing number of sequenced samples. In other words, all genes in the QZT samples used by the research institute have been detected.

Based on the relative abundance of microorganisms, principal component analysis (PCA) and Bray–Curtis distance cluster analysis of the samples during pile fermentation were performed. The results showed that, during the Qingzhuan tea pile fermentation process, the samples clustered into three groups: QZT1, QZT2, QZT3, and QZT4 in stage 1 belonged to group 1; QZT5, QZT6, and QZT7 in stage 2 to group 2; and QZT8, QZT9, and QZT10 in stage 3 to group 3. These clustering results were basically consistent with the three pile turning time points during pile fermentation. Stage 2 and stage 3 were clearly separated from stage 1, and stage 2 tended to move to stage 3. The fungal community was scattered in the early stage of pile fermentation and the microbial community of QZT6 in stage 2 exhibited the greatest changes, and then the microbial community in stage 2 tended to be consistent with that in stage 3 of pile fermentation (Figure 1).

#### 3.1.1. Changes in Microbial Abundance during Qingzhuan Tea Pile Process

In order to better demonstrate the differences in the microbial community structure during the Qingzhuan tea pile fermentation process, the relative abundances of fungi and bacteria in the macrogene data were analyzed. As shown in Figure 2A, at the kingdom level, fungi accounted for 93.51%~99.68%, indicating that fungi were the dominant microorganisms, and bacteria (0.32–6.49%) were auxiliary microorganisms in the pile fermentation process of Qingzhuan tea.

Subsequently, this study analyzed the changes in the relative abundance of microorganisms at kingdom levels. The results showed that, in the early stage of Qingzhuan tea pile fermentation (stage 1), the dominant microorganisms were *Lichtheimia* and *Helicocarpus* (Figure 2B). With the increasing temperature and the pile fermentation progression, the thermophilic fungi *Rasamsnia* and *Talaromyces* in *Trichocomaceae* family appeared, and their abundances were significantly increased in stage 3. Additionally, both *Aspergillus* and *Pichia* were expressed during the entire pile fermentation process, and, in the middle stage of pile fermentation (stage 2), the abundance of *Aspergillus* was significantly increased, while the abundance of *Pichia* remained relatively stable.

#### 3.1.2. Functional Annotation of Microorganisms during Qingzhuan Tea Fermentation Process

The non-redundant gene set in this study was aligned to the MicroNR library by Blastp and the LCA algorithm was used for species annotation. The proportions of annotated genera and phyla were 70.51% and 83.56%, respectively. Using DIAMOND software, a common functional database of non-redundant gene sets was annotated (e-value <= 10^−5^). A total of 13,759 (1.86%) ORFs were aligned to the CAZy database, 264,663 (35.72%) ORFs to the KEGG database, and 268,864 (36.29%) ORFs to the eggNOG database. Based on the resistance gene database (CARD), the non-redundant gene set was annotated (e-value <= 10–30) and 516 genes were aligned to the CARD database (Figure 3A).

The annotation results of KEGG metabolic pathways of microorganisms in the Qingzhuan tea pile process showed most of the microorganisms were mainly involved in biological metabolic pathways such as metabolism, cell process, and environmental information processing pathways. Among them, metabolism pathways were mainly related to carbohydrate metabolism, amino acid metabolism, energy metabolism, cofactor vitamin metabolism, fatty acid metabolism, and so on. In the cellular process pathways, microorganisms were mainly involved in transport and catabolism, cell growth, and death. Environment information processing biological metabolic pathways was mainly related to signal transduction.

#### 3.1.3. Identification of Differential Microorganisms in Qingzhuan Tea Pile Fermentation Process

In this study, linear discriminant analysis (LDA) in LEfSe was used to investigate the samples in the early (stage 1), middle (stage 2), and late (stage 3) three stages so as to identify differential microorganisms, and an LDA value > 4 and *p* < 0.05 were used as screening standards. A total of 110 differential microorganisms were identified, of which 7, 20, and 83 differential microorganisms were obtained from stage 1, stage 2, and stage 3, respectively (Figure 3B,C). The *Sphingomonas* genus contributed the most to stage 1 (the early stage of pile fermentation), *Lactobacillaceae* contributed the most to stage 2 (middle stage), and *Aspergillaceae* and *Trichocomaceae* contributed the most to stage 3 (late stage).

As shown in Figure 3C, among seven differential microorganisms significantly enriched in the early stage of pile fermentation (stage 1), six differential microorganisms were bacteria including four *Proteobacteria* phyla and two *Actinobacteria* phyla, except one fungus (*Mortierella* genus, *Mortierellaceae* family, *Mortierellales* order, and *Mucoromycota* phylum). In *Proteobacteria* microorganisms, there were two α-*Alphaproteobacteria* class, *Sphingomonadales* order, *Sphingomonadaceae* family, *Sphingomonas* genus, and one γ-*Gammaproteobacteria* class and *Xanthomonadales* order. The two *Actinomycetes* phyla belonged to the *Actinobacteria* class.

Twenty differential microorganisms significantly enriched in the middle stage of pile fermentation (stage 2) included 14 bacteria, 4 fungi, and 2 viruses. The 14 differential bacteria significantly enriched in stage 2 were composed of 6 *Firmicutes* phyla and 7 *Proteobacteria* phyla as well as bacterial communities at various levels of order, family, and genera contained therein: *Firmicutes* and its *Lactobacillus* genus (5) and *Weissella* genus (1), *Proteobacteria* and its *Enterobacteriaceae* family (4) and *Acetobacter* genera (3). The 4 differential fungal communities significantly enriched in stage 2 included *Mucoromycota* phylum (1), *Ascomycota* phyla (3), and fungal communities at various levels of the order, family, and genera contained therein: *Ascomycota* phylum and its *Peltigerales order* (2) and *Pichia* genus (1).

The differential microorganisms significantly enriched in stage 3 (the late stage of pile fermentation) included 15 *Actinobacteria* bacterial classes and 68 *Ascomycota* fungal phyla as well as the communities at various levels of order, family, and genus contained therein. Differential microorganisms in bacteria included *Saccharopolyspora* genera (8), *Actinopolysporaceae* families (4), and *Streptomycetales* orders (2). Except for 2 *Agaricomycetes* classes from the *Basidiomycota* phylum, the other 68 significantly enriched differential fungal microorganisms were all from the *Ascomycota* phylum, including *Eurotiomycetes* classes (28), *Lecanoromycetes* classes (10), *Sordariomycetes* classes (9), *Dothideomycetes* classes (8), *Leotiomycetes* classes (7), and *Pezizomycetes* classes (4). The *Eurotiomycetes* class with the most differential microorganisms included 9 *Onygenales* orders and 19 *Eurotiales* orders. The *Aspergillaceae* family (8 species), *Trichocomaceae* family (5 species), and *Elaphomycetaceae* family (3 species) were the main microorganisms in the *Eurotiales* order. The species *Aspergillus niger*, *Aspergillus_wentii*, *Aspergillus_turcosus*, *Aspergillus_clavatus*, *Penicillium_subrubescens*, and *Penicilliopsis_zonata* in the *Aspergillaceae* family, and *S-Rasamsonia_emersonii*, *Talaromyces_marneffei*, and *Talaromyces_verruculosus* in the *Trichocomaceae* family were found.

### 3.2. Analysis of Key Microorganisms Affecting Quality and Their Metabolites

Further, we also conducted metabolomic analysis on the same batch of Qingzhuan tea processing samples (Figure 4) and screened a total of 53 metabolites (of which 26 were annotated) with VIP > 1.5 and *p* < 0.05 as screening thresholds. Most of the screened compounds were enriched in amino acid and polyphenol metabolic pathways. With the pile fermentation progression, the contents of polyphenolic metabolites were decreased to varying degrees mainly including amino acids such as theanine (ID, 274), chlorogenic acid (ID, 756), GCG (ID, 491), puerin of the two types of B-ring dihydroxyl (ID, 733, ID, 582), galloyl puerin B-ring trihydroxyl (ID, 597), and others. However, the contents of tannin (ID, 688) and amino acid analogs (ID, 889) in the polyphenol metabolic pathway were increased to varying degrees with the progression of pile fermentation.

Proctor’s analysis examined the correlation between the microorganism and metabolite profiles of Qingzhuan tea. As shown in Figure 5, the rectangle shape in the Proctor’s analysis diagram represents the metabolic profile of green brick tea, while the blue arrow points to the microorganism. According to the characteristics of Proctor’s analysis, the correlation between the two matrices and the 999 replacement tests (M^2^ = 0.52, *p* < 0.05) revealed that there was a significant correlation between microorganisms and the metabolic profile of Qingzhuan tea. Correlation analysis was conducted between 53 metabolites with VIP > 1.5 identified by Qingzhuan tea metabolomics method and 106 microorganisms screened by metagenomic LEfSe analysis (Figure 5). The results showed that the metabolites associated with microorganisms were mainly derived from the polyphenol metabolic pathway and amino acid metabolic pathway. The obtained 49 VIP compounds (26 annotated) were significantly correlated with 96 target microorganisms. The correlation between VIP compounds and eight target microorganism strains (including two strains of *Sphingomonadaceae* family and two strains of *Glomeromycetes* classes) exhibited the pattern opposite to that between VIP compounds and the other 88 target microorganisms, which might be closely related to their different growth characteristics and nutritional requirements. Subsequent analysis found that, during the process of Qingzhuan tea pile fermentation, the relative expression patterns of 23 metabolites (11 annotated) were the same as those of most screened microorganisms, while the relative expression patterns of 28 metabolites were opposite to those of most screened microorganisms.

The metabolites with the same expression pattern as the screened microorganisms were mainly from polyphenol metabolism-related pathways (8/11), chlorogenic acid (ID, 756), GCG (ID, 491), phenolic acids (ID, 567), phenolic amides (ID, 694), two kinds of B-ring dihydroxy pu-er (ID, 733, ID, 582), B-ring tri-hydroxyl galloyl pu-er (ID, 597), and other compounds. Three other amide metabolites (ID, 484; ID, 496; and ID, 557) and the target microorganism exhibited the same relative expression pattern. The metabolites whose expression patterns were opposite to those of screened microorganisms mainly involved four amide compounds composed of (ID, 644), (ID, 295), fatty amides (ID, 435), and cyclic peptides (ID, 209); four amino acids including theanine (ID, 274), glutamine (ID, 231), amino acid analogs (ID, 476), and (ID, 889); two polyphenol metabolites including tannins (ID, 688) and phenolic acid (ID, 32), and two disaccharide metabolites, (ID, 463) and (ID, 289).

The metagenomic analysis showed that the microorganism with the largest contribution to stage 1 was the *Sphingomonas* genus, which mainly showed a significant positive correlation with amino acids (ID,889; ID, 231; ID, 274; ID, 280) and B-ring dihydroxy puertin (ID, 582) metabolism compounds, and the *Aspergillaceae* family and *Trichocomaceae* family contributed greatly to stage 3. Among the *Aspergillaceae* family, *Aspergillus_fumigatus* was only significantly correlated to compound (ID, 694), while *Aspergillus_tubingensis* and *Aspergillus_niger* were significantly correlated with (ID, 694). *Aspergillaceae* family members such as *Paecilomyces_Penicilliopsis*, *Aspergillus_wentii*, and *Aspergillus_clavatus*, and the *Trichocomaceae* family showed correlations with almost all metabolites. Among them, GCG showed a significant positive correlation with the *Ajellomycetaceae* family of the *Eurotiales* order, *Aspergillus_wentii* of the *Aspergillaceae* order, and *Verticillium_dahliae* of the *Sordariomycetes* class. However, purine and catechin showed a positive correlation with most of the target microorganisms. The correlation pattern between microorganisms and GCG or B-ring trihydroxy galloyl puerlin (ID, 597) was completely opposite to that between microorganisms and all identified amino acids. These completely opposite correlation patterns might regulate the carbon/nitrogen balance in tea leaves to some extent during the pile fermentation process of Qingzhuan tea. Amino acids and flavonoids represent nitrogen-containing and carbon-rich metabolites, respectively, and thus their metabolisms are regulated by the carbon/nitrogen balance. Glu and ethylamine are the precursors synthesized by theanine, while phenylalanine is the starting unit of flavonoid metabolism. All the carbon skeletons used for synthesizing Glu, ethylamine, and Phe are derived from the intermediates (phosphoenol-pyruvate, pyruvate, and 2-oxoglutarate) produced from glycolysis and the tricarboxylic acid (TCA) cycle. The shared carbon backbones may be a key factor regulating the compound flow between the amino acid pathway and the flavonoid pathway [30,31].

## 4. Discussion

During the fermentation process of Qingzhuan tea, the microbial community plays an important role in the quality formation and material transformation process. Previous studies have shown that the enzymatic reaction during the pile fermentation process is mainly responsible for the reduction in chemical components [21,28]. Amino acids, caffeine, theobromine, and theophylline are the main nitrogen-containing compounds in dark raw tea, but alkaloids, such as heterocyclic compounds, are difficult for microorganisms to utilize. Therefore, amino acids are the most important microbial nitrogen sources in the fermentation process of dark tea and they are also an important factor affecting the structure of microorganisms. By the traditional separation and culture method, Zhang et al. found that, in the pile fermentation process of Qingzhuan tea, the number of bacteria was the largest, followed by *Actinomycetes* and molds, and yeast was the least [32]. Wang et al. isolated and identified the fungus *Mucor racemosa* from the pile fermentation process of Qingzhuan tea and simulated fermentation, and found that this strain was an important strain involved in pile fermentation process of Qingzhuan tea. The evolution of the microbial community structure may be due to changes in metabolite contents during the pile fermentation [33].

Our results showed that fungi were more important than bacteria in the pile fermentation of dark tea, which was similar to the previous report on Fuzhuan tea [2,21]. *Aspergillus niger*, *Aspergillum gloucus*, and *Penicillium* were identified as the dominant strains in the fermentation process of Pu’er tea [18,19], Fuzhuan tea [5], and Qingzhuan tea [7]. *Talaromyces* (three species) and *Rasamsonia* (two species) of the *Trichocomaceae* family were also identified as differential microorganisms in this study, and they were detected in Liubao tea and Puer tea [34,35], but their role in fermentation remains unclear. *Talaromyces* is a thermotolerant genus that mostly grows and reproduces in high-temperature, high-humidity, aerobic organic matter clumps and it can secrete high temperature-resistant hydrolytic enzymes. *Rasamsonia* is a new genus including thermotolerant and thermophilic *Talaromyces* and *Geosmithia* species and it has a strong activity of decomposing cellulose at about 50 °C, and the rate of degrading cellulose of thermophilic *Sporotrichum* is five times that of the normal temperature strain *Trichoderma* [34,35].

Molds are the most common microorganisms found in the fermentation process of Pu-er tea. As the dominant strain, *Aspergillus* can secrete α-amylase, glucoamylase, cellulase, pectinase, and other proteases, and it plays an important role in decomposing proteins and lipids, and producing the unique flavor of tea [18,19]. In this study, *Aspergillus niger*, *Aspergillum gloucus*, *Penicillium* of the *Aspergillaceae* family, and *Talaromyces* and *Rasamsonia* of the *Trichocomaceae* family were identified as the dominant strains in the process of pile fermentation of Qingzhuan tea, and they were important participants in the formation of the sensory qualities of Qingzhuan tea such as the unique flavor and taste. However, a series of scientific issues such as how these dominant microorganisms, the *Aspergillus* family and the *Trichocomaceae* family, interact with the tea components and how they participate in the formation of quality compounds need to be further explored.

One previous study has shown that, under the action of fungal fermentation, Puerins I~VIII formed from the condensation transformation of catechin and theanine were detected in Pu’er tea and Fuzhuan tea, and their (-)-epicatechin or (-)-epigallocatechin parts were replaced by N-ethylpyrrolidone [36]. Two puerins, 402.155 666 1 (ID, 733) and 402.155 722 1 (ID, 582), were identified, which were positively correlated with most of the target microorganisms, while amino acids (especially theanine and glutamine) were significantly negatively correlated with microorganisms. Microorganisms such as *Trichocomaceae*, *Elaphomycetaceae*, *Aspergillaceae* in the *Eurotiomycetes* family and *Eurotiales* order were the main contributors. Puerin extracts from Pu-er tea have been confirmed to significantly reduce the levels of blood sugar, total cholesterol, and triglycerides in plasma, and their hypoglycemic effect might be derived from their inhibition of α-glycosidase activity [37]. There are no reports on the effects of the types and structures of puerins in Qingzhuan tea on tea quality formation and biological functions. Further research is needed to explore the contribution of puerins to the characteristic quality formation of Qingzhuan tea, its health efficacy in lowering blood sugar and lipids, and its influence on microorganisms.

## 5. Conclusions

In the early stage (stage 1) and the middle stage (stage 2) of Qingzhuan tea pile fermentation, the dominant microorganisms were bacteria. In the late stage (stage 3) of pile fermentation, the dominant fungi were mainly *Aspergillaceae*, *Trichocomaceae*, and *Elaphomycetaceae*. The *Aspergillaceae* family included *Aspergillus niger*, *Aspergillum gloucus*, *Aspergillus wentii*, *Aspergillus turcosus*, *Aspergillus clavatus*, and *Penicillium*. Amino acids and polyphenol metabolites with relatively simple structures showed a significant negative correlation with target microorganisms, while the complexes with complicated structures such as catechin–amino acid and catechin–purine showed a significant positive correlation with target microorganisms. *Aspergillus niger, Aspergillums gloucus*, and *Penicillium* in the *Aspergillaceae* family, and *Talaromyces* and *Rasamsonia emersonii* in the *Trichocomaceae* family were the key microorganisms for the characteristic quality formation of Qingzhuan tea.

## Figures and Tables

**Figure 1 foods-12-03537-f001:**
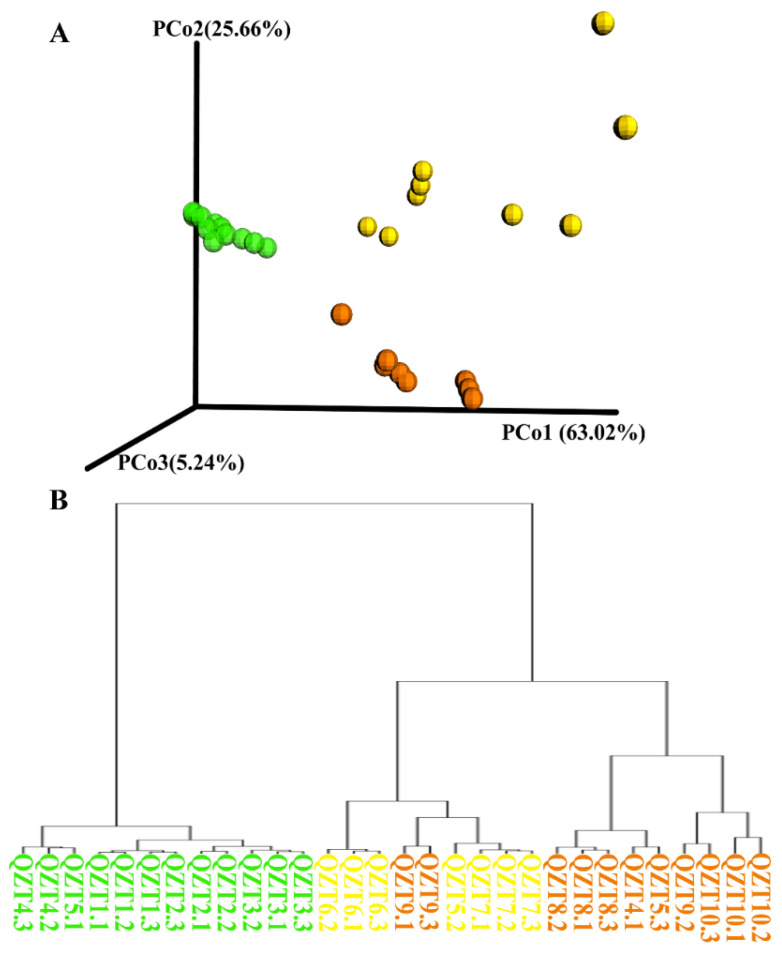
PCA analysis and cluster analysis based on Bray–Curtis distance in Qingzhuan tea pile fermentation process. (**A**) Different color dots represent different pile fermentation stages. The green dots represent stage 1, the yellow dots indicate stage 2, and the orange dots denote stage 3. (**B**) Samples of the cluster analysis based on Bray–Curtis distance are labeled with different colors and offer information about where the various stages of Qingzhuan tea fermentation are located. The color annotation is the same as that in (**A**). Metagenomic analyses were conducted with three biological replicates.

**Figure 2 foods-12-03537-f002:**
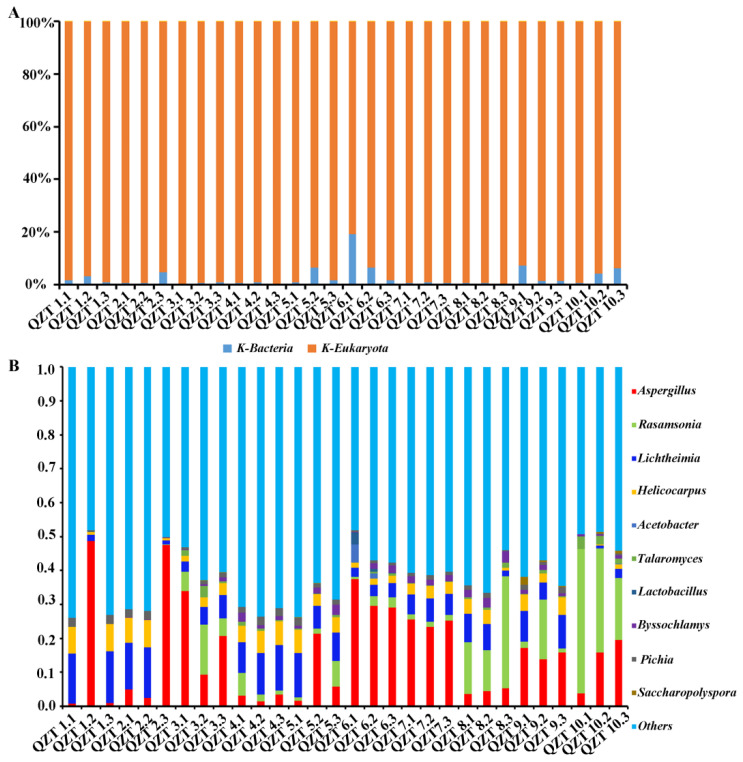
Relative abundance changes in microorganisms during Qingzhuan tea pile fermentation. (**A**) Bacteria and fungi. (**B**) Top 10 microorganisms at genus level. Metagenomic analyses were conducted with three biological replicates.

**Figure 3 foods-12-03537-f003:**
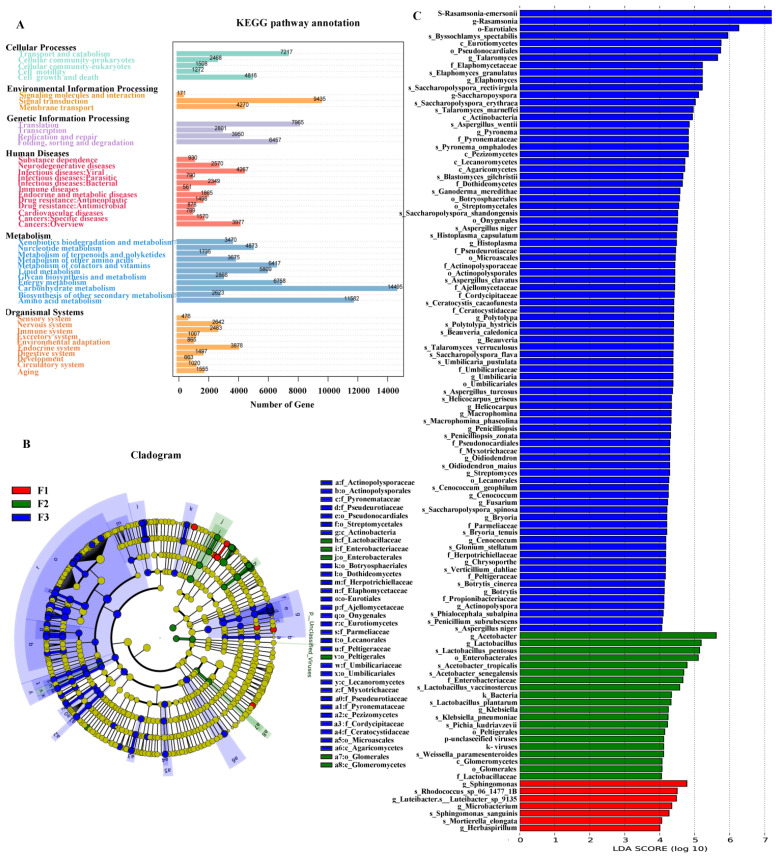
Statistical diagram of number of KEGG-annotated genes and LEfSe analysis of differential microorganisms in Qingzhuan tea pile fermentation. (**A**) Statistical diagram of number of KEGG-annotated genes. (**B**) The cladogram plot of LEfSe analysis. (**C**) Microorganism differences among the three groups were identified with a LEfSe analysis with LDA score threshold > 4.0.

**Figure 4 foods-12-03537-f004:**
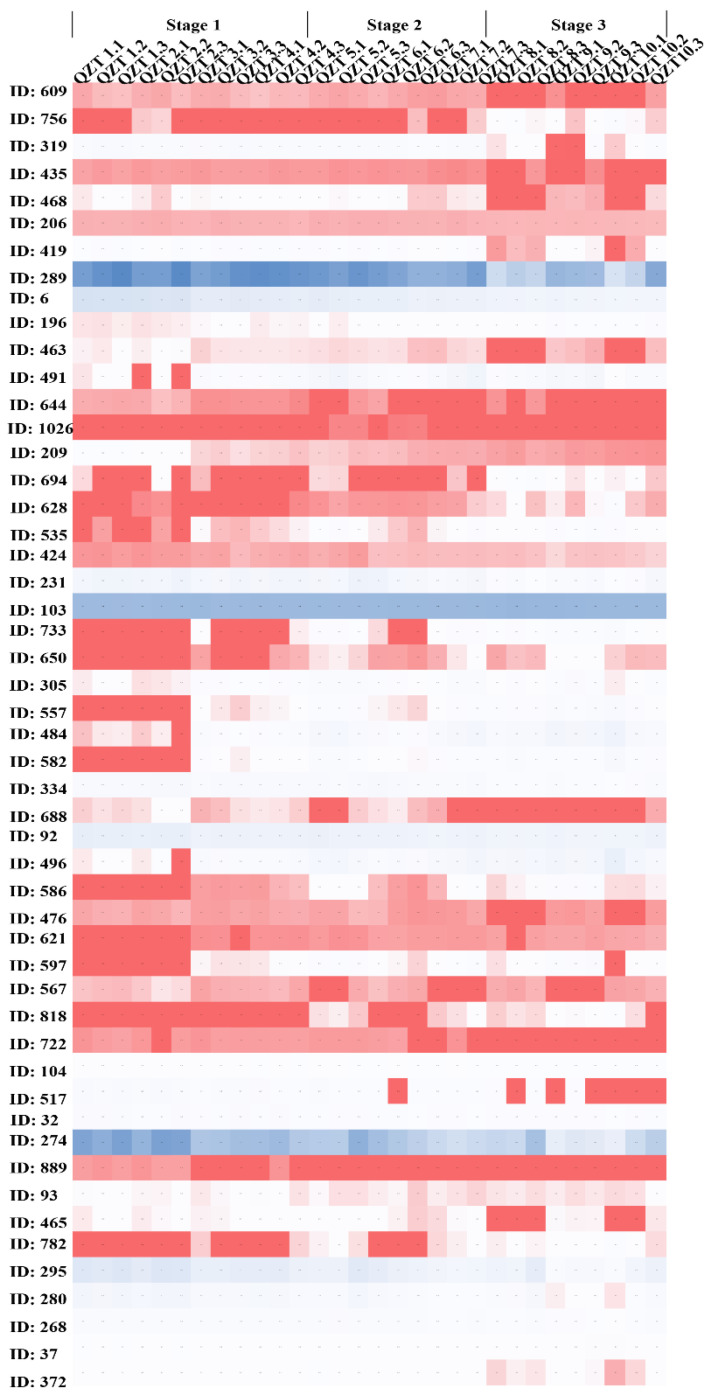
Heatmap of VIP compounds in pile fermentation of Qingzhuan tea. The metabolic analyses were conducted with three biological replicates.

**Figure 5 foods-12-03537-f005:**
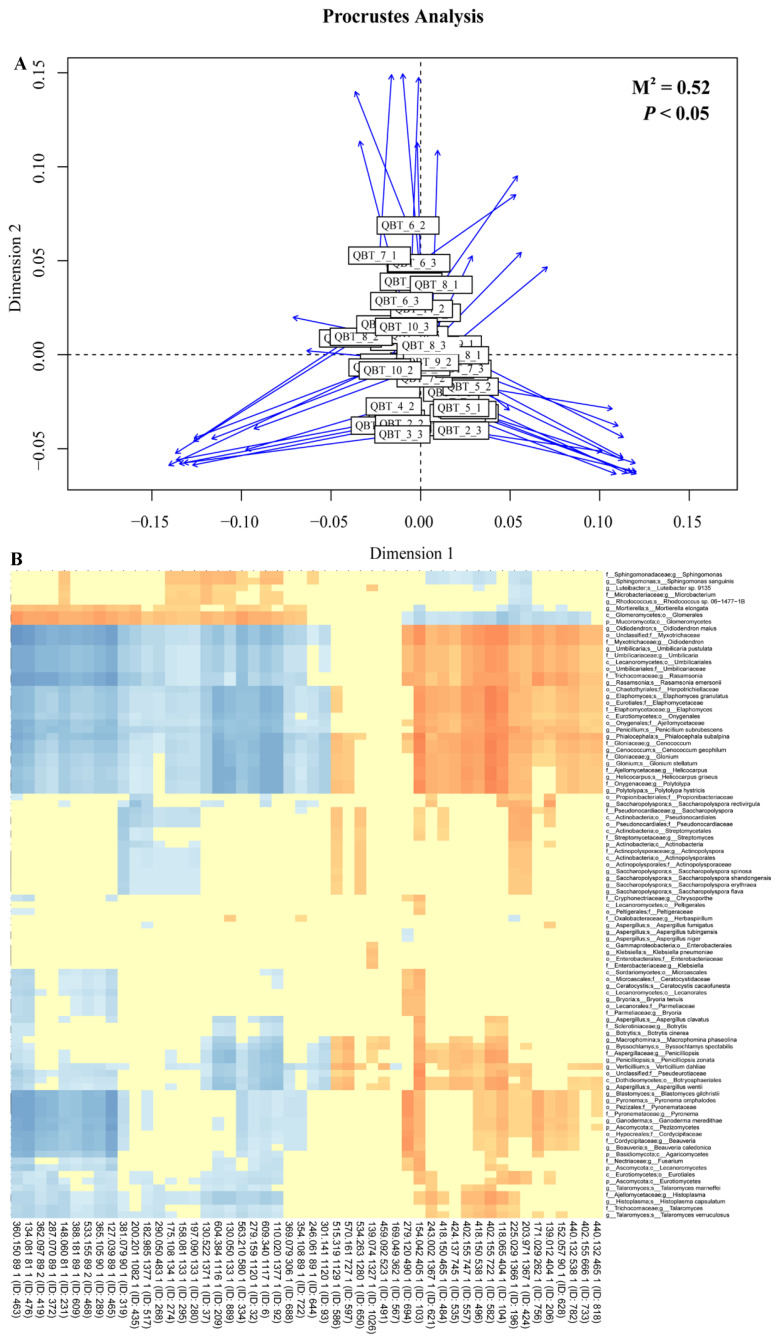
Matrix correlation analysis and correlation analysis between target microorganisms and differential VIP metabolites screened during the fermentation process of Qingzhuan tea. (**A**) Matrix correlation analysis of differential VIP metabolites and microorganisms. (**B**) Correlation analysis between target microorganisms and differential VIP metabolites.

## Data Availability

The data used to support the findings of this study can be made available by the corresponding author upon request.

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
