# Peer review of "Microbial Diversity and Characteristic Quality Formation of Qingzhuan Tea as Revealed by Metagenomic and Metabolomic Analysis during Pile Fermentation"

_foods, 2023, doi:10.3390/foods12193537_

Round 1

Reviewer 2 Report

This important study analyzed the changes in the microbial community structure during the fermentation of Qingzhuan tea and investigated their correlation with the formation of quality compounds using metagenomic and metabolomic techniques. In order to improve quality of this manuscript, the following issues must be addressed.

In Section 3.12, the authors reported that, as shown in Figure 2A, fungi accounted for 93.51 % 99.68 % of the pile fermentation process of Qingzhuan tea, while bacteria accounted for 0.32 % to 6.49 %. During the Qingzhuan tea pile fermentation process, it is crucial to explain when (at what stage) these amounts were determined.

The legend for Figure 1 should illustrate what the different colors represent and where the various stages of Qingzhuan tea fermentation are located.

The authors must also elucidate the significance of the different numbering (.1,.2, and.3) in Figures 1 and 2. Currently, they are perplexing.

Section3.1.3: The stages at which relative abundance variations of microorganisms were evaluated during Qingzhuan tea pile fermentation must be specified.

Figure 3 caption: Figure 3 a and b are both labeled Statistical diagram of the number of KEGG-annotated genes. This is ambiguous and should be corrected

Chapter 3.1.4: The acetobacter and S-Rasamsonia emersonii genera, which substantially contributed the most to Stages 2 and 3, respectively, are not mentioned. This section appears to contain a contradiction. It is essential to revisit, verify, and confirm the Identification of these distinct microorganisms as specified in section 3.1.4.

Sensory analysis often follows similar investigations. It is remarkable that the authors did not perform Sensory analysis at various fermentation stages. This could have enhanced the omics (metagenomic and metabolomic analyses) and the tea's qualitative characteristics.

 The authors should also attempt to highlight the novelty of this study.

Legends of the supplementary figures must be provided

There are too many spacing errors in the manuscript. This must be checked. I highlighted some in yellow.

Grammar must also be checked 

There are too many spacing errors in the manuscript. This must be checked. I highlighted some in yellow.

Grammar must also be checked 

Round 2

Reviewer 2 Report

The authors have addressed all my concerns. Thanks

English quality seems fine
